# *Toxoplasma* GRA15 and GRA24 are important activators of the host innate immune response in the absence of TLR11

**Debanjan Mukhopadhyay**, **David Arranz-Solís**, **Jeroen P. J. Saeij***

Department of Pathology, Microbiology and Immunology, School of Veterinary Medicine, University of California Davis, Davis, California, United States of America

* jsaeij@ucdavis.edu

**Data Availability Statement:** All relevant data are within the manuscript and its Supporting Information files.

## Abstract

The murine innate immune response against *Toxoplasma gondii* is predominated by the interaction of TLR11/12 with *Toxoplasma* profilin. However, mice lacking *Tlr11* or humans, who do not have functional TLR11 or TLR12, still elicit a strong innate immune response upon *Toxoplasma* infection. The parasite factors that determine this immune response are largely unknown. Herein, we investigated two dense granule proteins (GRAs) secreted by *Toxoplasma*, GRA15 and GRA24, for their role in stimulating the innate immune response in *Tlr11$^{-/-}$* mice and in human cells, which naturally lack TLR11/TLR12. Our results show that GRA15 and GRA24 synergistically shape the early immune response and parasite virulence in *Tlr11$^{-/-}$* mice, with GRA15 as the predominant effector. Nevertheless, acute virulence in *Tlr11$^{-/-}$* mice is still dominated by allelic combinations of *ROP18* and *ROP5*, which are effectors that determine evasion of the immunity-related GTPases. In human macrophages, GRA15 and GRA24 play a major role in the induction of IL12, IL18 and IL1β secretion. We further show that GRA15/GRA24-mediated IL12, IL18 and IL1β secretion activates IFNγ secretion by peripheral blood mononuclear cells (PBMCs), which controls *Toxoplasma* proliferation. Taken together, our study demonstrates the important role of GRA15 and GRA24 in activating the innate immune response in hosts lacking TLR11.

## Author summary

In mice, the early immune response against *Toxoplasma* is dominated by TLR11-mediated release of IL12, which subsequently induces protective IFNγ. Here we show that in *Tlr11$^{-/-}$* mice and in human cells, which do not have TLR11, the *Toxoplasma* GRA15 and GRA24 effectors play an important role in induction of IL12, IL18 and IL1β, and thus in the subsequent protective IFNγ secretion.

## Introduction

*Toxoplasma gondii* is an obligate intracellular parasite capable of infecting any nucleated cell of any warm-blooded animal, including humans. It can cause lifelong persistent infections by

**Funding:** This study was supported by the National Institutes of Health (R01-AI080621) awarded to J. P.J.S. DM was supported by the American Heart Association Post-doctoral fellowship (18POST34030036). The funders had no role in study design, data collection and analysis, decision to publish, or preparation of the manuscript.

**Competing interests:** The authors have declared that no competing interests exist.

forming semi-dormant cysts in muscles and the brain [1–3]. *Toxoplasma* resides within a non-fusogenic vacuole called the parasitophorous vacuole (PV), which is separated from the host cell cytosol by the PV membrane (PVM), preventing the parasite from being recognized by the host innate immune system. However, the cytokine interferon-gamma (IFNγ) activates effector mechanisms that can mediate the elimination of *Toxoplasma*. Inflammatory cytokines produced by macrophages and dendritic cells (DCs) in response to Toll-like receptor (TLR) recognition of conserved pathogen associated molecular patterns (PAMPs) are important for subsequent production of IFNγ. For example, in mice the *Toxoplasma* actin-binding protein profilin is recognized by a heterodimer of TLR11/12 that is located in the endosome, inducing a signaling cascade leading to the production of interleukin (IL)12 by DCs [4–6]. IL12 in turn activates Natural Killer (NK) and T cells to secrete IFNγ, which can trigger a variety of toxoplasmacidal mechanisms [7, 8]. In mice, IFNγ-induced immunity related GTPases (IRGs) that can coat and vesiculate the PVM, and ultimately destroy the parasite inside, play a dominant role in resistance to *Toxoplasma* [9–11].

Innate immunity can also be activated by specific cytosolic receptors (often nucleotide-binding domain and leucine-rich repeat-containing receptors or NLRs) as a part of a multi-protein complex called the inflammasome [12]. In mice, *Toxoplasma* can activate the NLRP1 and NLRP3 inflammasomes [13], leading to IL1β/IL18 production, which together with IL12 can enhance IFNγ secretion and thereby contribute to host resistance against *Toxoplasma* [14, 15]. However, inflammasome activation in *Tlr11*$^{-/-}$ mice can also induce a pathological inflammatory response [16].

Human sensing and killing of *Toxoplasma* differs from mice, as humans lack functional TLR11/12 and do not have IRGs [17]. It was recently shown that the alarmin S100A11 secreted from infected monocytes or fibroblasts can shape the human immune response through secretion of the chemokine ligand 2 (CCL2) [18]. In addition, cytosolic recognition of *Toxoplasma* in human monocytes was shown to partly rely on the NLRP1 and NLRP3 inflammasome, resulting in secretion of IL1β [19, 20]. Furthermore, guanylate binding protein (GBP)1 facilitates disruption of the PV in IFNγ-stimulated human macrophages, which causes release of parasite nucleic acids that can activate cytosolic absent in melanoma 2 (AIM2) and caspase 8-dependent apoptosis [21]. This could be another potential route of immune activation as recognition of cytosolic nucleic acids induces the type I interferon pathway.

*Toxoplasma* can counteract the host immune response by secreting effector proteins, ROPs and GRAs, into the host cell from specialized secretory organelles called rhoptries and dense granules, respectively [22, 23]. In Europe and North America, strains belonging to four different *Toxoplasma* clonal lineages (types I, II, III and XII) are commonly isolated in animals and humans, although most infections are caused by type II strains [24–26]. In mice, strain differences in virulence and modulation of host cell signaling are largely due to polymorphisms in ROPs and GRAs. For example, ROP18 (a secreted kinase) and ROP5 (a pseudokinase) determine strain differences in virulence in mice by cooperatively blocking the IFNγ-induced IRGs [10, 27–29]. Several GRA proteins are localized on the PVM and can modulate the host immune response [30–32]. For instance, GRA15 from type II strains activates the NFκB pathway, leading to macrophage production of inflammatory cytokines such as IL12 and IL1β [31, 33]. Other GRAs are secreted beyond the PVM, where they can modulate host cell signaling pathways [22, 34–36]. For example, GRA24 binds to p38α MAPK leading to its autophosphorylation and constitutive activation [35]. Together, GRA15 and GRA24 drive the classical activation of macrophages (M1) via the activation of NFκB and p38 MAPK [35, 37, 38]. By contrast, the polymorphic kinase ROP16 from type I and type III strains drives the alternative activation of macrophages (M2) via the phosphorylation of the STAT6 and STAT3 transcription factors [37, 39, 40]. It is likely that the deliberate activation of the immune response by

*Toxoplasma* effectors is a strategy to limit its virulence thereby promoting the survival of its host and the formation of tissue cysts, which are the only stages in the intermediate host that are orally infectious.

Given the large impact of GRA15 and GRA24 on macrophage gene expression and production of IL12 and IL1β, it seems surprising that their *in vivo* effect on parasite virulence is relatively minor [31, 35]. Mice infected with type II Δ*gra15* or Δ*gra24* parasites had elevated parasite numbers early after infection, but as the infection progressed, parasite burden and host susceptibility were no different from those following wild-type type II strain infections [31, 35, 41]. Increased type II Δ*gra15* early parasite burdens were associated with decreased IL12 and IFNγ levels 2 days after infection [31]. Similar results were obtained after infection of mice with the Δ*gra24* strain [35]. Possibly, the effects of GRA15 and GRA24 in these studies were masked by profilin: as the infection progresses and parasites lyse out of host cells, PAMPS, such as profilin, are released and activate TLR11/12. At this stage, IL12/IL1β and subsequent IFNγ production are probably no longer dependent on GRA15 and GRA24. However, humans and many animals do not have functional TLR11/12 or IFNγ-inducible IRGs [17], and therefore *Toxoplasma* virulence of a particular strain in mice might not correlate with virulence in other species. In *Tlr11*[-/-] mice, neutrophils are the main producers of IFNγ with a minor role of NK and T cells [42]. The production of IFNγ by neutrophils is dependent on IL1β and TNFα, but not on IL12 [42]. In addition, IL18 secreted upon inflammasome activation plays a key role in the IFNγ response from CD4[+] T cells and the subsequent disease outcome in *Tlr11*[-/-] mice [43]. Thus, GRA15 and GRA24, by inducing IL1β, IL18, TNFα and IL12, might play an important role in the production of IFNγ in hosts lacking TLR11. Herein, we tested this hypothesis by infecting *Tlr11*[-/-] mice with wild-type, Δ*gra15*, Δ*gra24* and Δ*gra15/24* parasites. Our data indicate that although parasites that do not express GRA15 and/ or GRA24 induced significantly less inflammatory cytokines, a significant increase in virulence compared to wild-type was only observed after subcutaneous infection, likely because *Tlr11*[-/-] mice were already extremely susceptible to wild-type *Toxoplasma* infections. We further show that in *Tlr11*[-/-] mice IRG-mediated killing of *Toxoplasma* is likely still the major mechanism of resistance, as parasites that express avirulent ROP5 and ROP18 were completely avirulent in these mice. In human THP1-derived macrophages and PBMCs, GRA15 and GRA24 determined the induction of inflammatory cytokines and thereby had a large effect on parasite proliferation. Thus, in the absence of TLR11, GRA15 and GRA24 are the major parasite effectors that activate the innate immune response and it is likely that in humans they determine parasite virulence.

## Results

### GRA15 and GRA24 regulate murine macrophage function *in vitro*

Synthesis of IL12 by *Toxoplasma*-infected macrophages was previously shown to be dependent on p38 MAPK and NFκB activation [44, 45]. Indeed, deletion of either *GRA15* or *GRA24* significantly reduced macrophage IL12/IL23p40 production, consistent with their activation of the NFκB and p38 MAPK pathway, respectively [31, 35]. It was previously shown that GRA24 from both type I RH and type II Pru strains activates the p38 MAPK pathway. However, the GRA24-dependent transcriptional changes in murine macrophages are much more pronounced after infection with type II strains compared to type I strains [35]. For example, without GRA24 the Pru induction of inflammatory cytokines, including IL12/IL23p40, is significantly affected, while GRA24 only has a minor effect on the modulation of these cytokines by RH. It is possible that many of the GRA24-mediated transcriptional changes are dependent on GRA15, as RH does not express a functional GRA15 [31]. To more directly test whether

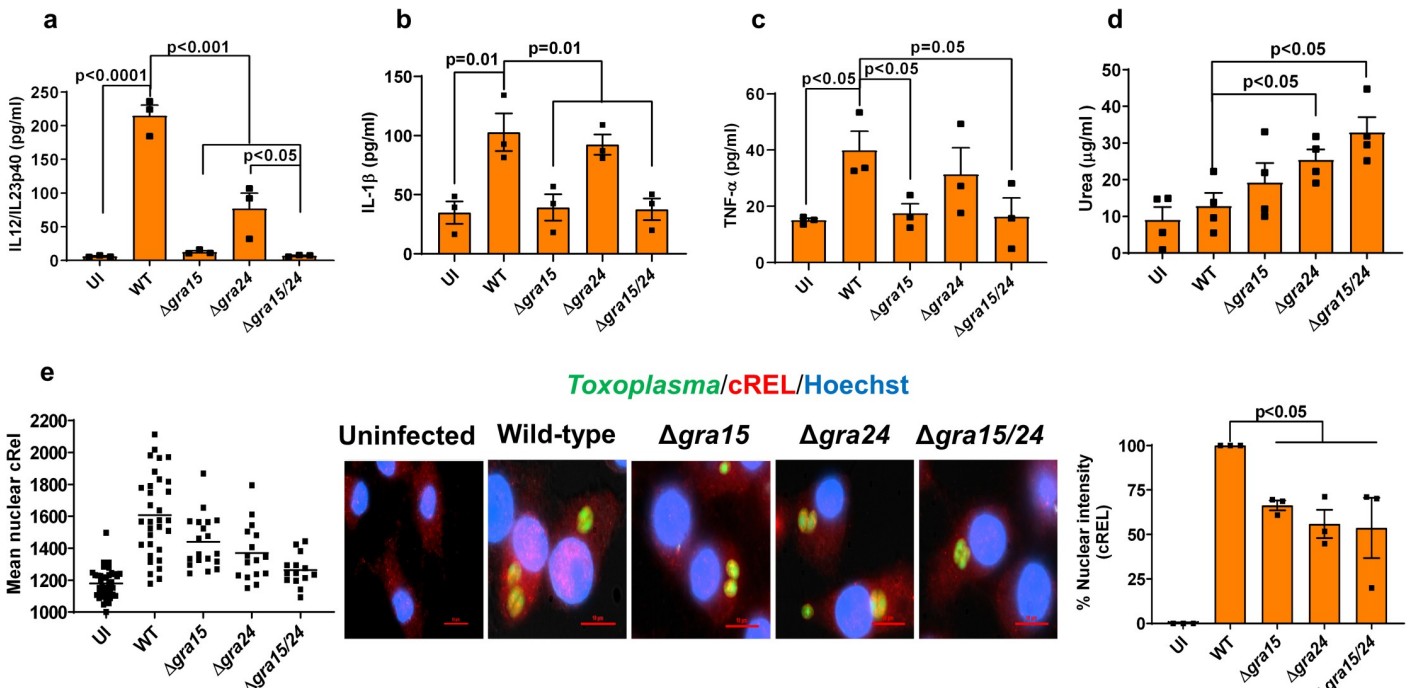

**Fig 1. GRA15 and GRA24 activate the *in vitro* macrophage response.** BMDMs were infected with indicated *Toxoplasma* strains for 24 h and IL12/IL23p40 (**a**), IL1β (**b**) and TNFα (**c**) were measured in the supernatant. Experiments were done 3 times. Arginase activity was measured (**d**) from RAW 264.7 macrophages infected with indicated strains 24 h p.i. Nuclear translocation of the cREL subunit of NFκB (**e**) was quantified from infected RAW 264.7 macrophages 18 h p.i. with indicated strains. In each experiment at least 15 cells were quantified as shown in the graph (left) and experiment was done 3 times (right panel graph). A representative image for each group is shown in the middle panel. Scale bar is 10 μm. Each dot represents the mean of 3 technical replicates from individual experiments, except for the scatter diagram in (**e**). Statistical analysis was done by two sample student's t test for figures a-e. Data are represented as mean ± standard error of the mean (SEM).

GRA15 and GRA24 have any synergistic or additive effect on macrophages, we generated *gra24* single knockout and *gra15/gra24* double knockout parasites in the type II Pru strain (**S1 Fig**). We infected murine bone marrow derived macrophages (BMDMs) with Pru wild-type, Δ*gra15* [31], Δ*gra24* and Δ*gra15/24* parasites for 24 h and measured the release of IL12/IL23p40, IL1β and TNFα, which are cytokines often used as M1 macrophage polarization markers [38]. BMDMs infected with Δ*gra15* or Δ*gra15/24* parasites secreted significantly less IL12/IL23p40, IL1β and TNFα (**Fig 1A–1C**) compared to wild-type infected BMDMs. Deletion of *GRA24* significantly impaired the release of IL12/IL23p40 but had no effect on IL1β and TNFα (**Fig 1A–1C**). We also measured arginase activity (a marker for M2 macrophages) in macrophages infected with Δ*gra15*, Δ*gra24* or Δ*gra15/24* parasites. Δ*gra24* and Δ*gra15/24*, but not wild-type and Δ*gra15* parasites, induced significant arginase activity in macrophages (**Fig 1D**). The transcription of the p40 subunit of IL12 in mice is primarily dependent on cREL and moderately on the NFκB p65 subunit [46]. Macrophages infected with wild-type parasites contained significantly more nuclear (activated) cREL compared to macrophages infected with Δ*gra15*, Δ*gra24* or Δ*gra15/24* parasites (**Fig 1E**). Similar results were obtained in human foreskin fibroblasts (HFFs) (**S3 Fig**). Nuclear translocation of the NFκB p65 subunit in macrophages and HFFs was significantly reduced after infection with Δ*gra15* or Δ*gra15/24* parasites but not after infection with Δ*gra24* parasites (**S1E and S1F Fig**). Thus, our results show that while GRA15 is required for the secretion of IL1β and TNFα from BMDM, both GRA15 and GRA24 are required for the secretion of IL12/IL23p40.

## Deletion of *GRA15* and *GRA24* affects *in vivo* parasite growth and cytokine production in *Tlr11*<sup>-/-</sup> mice

It was previously shown that the intraperitoneal parasite load in C57BL/6 mice intraperitoneally (i.p.) infected with either Δ*gra15* or Δ*gra24* was higher compared to wild-type parasites, but the mortality of the mice was not significantly different [31, 35]. To determine if *Toxoplasma* profilin-mediated activation of TLR11 might have masked the effect of GRA15 and GRA24 we investigated parasite burden and mortality in the *Tlr11*<sup>-/-</sup> mice. *Tlr11*<sup>-/-</sup> mice i.p. infected with Δ*gra15*, Δ*gra24* or Δ*gra15/24* parasites had a significantly larger peritoneal parasite load 3 days p.i. compared to wild-type infected mice, with Δ*gra15*-infected mice having the highest parasite load (**Fig 2A**). On day 1 p.i., the level of IL12/IL23p40, IFNγ, IL1β or TNFα in the serum was not higher than in uninfected mice. It was recently shown that in *Tlr11*<sup>-/-</sup> mice IL18 is necessary and sufficient for induction of IFNγ production [43]. The IL18 level was significantly increased in mice infected with wild-type parasites (**Fig 2B**) while mice infected with Δ*gra15* or Δ*gra24* parasites had significantly lower serum IL18 levels on day 1 p.i. compared to wild-type parasite infected mice, which was even further reduced in mice infected with Δ*gra15/24* parasites. On day 1 p.i. IFNγ, IL18, IL1β or TNFα were not detected in the peritoneal fluid but the IL12/IL23p40 levels were significantly higher in mice infected with wild-type parasite compared to uninfected mice (**S4A Fig**). Mice infected with Δ*gra15/24* parasites contained significantly lower IL12/IL23p40 levels compared to wild-type parasite infected mice (**S4A Fig**). On day 3 p.i., the IL18 level in serum still remained significantly higher in wild-type parasite infected mice, while Δ*gra15*, Δ*gra24* or Δ*gra15/24* parasites elicited significantly lower IL18 levels (**Fig 2C**). Although IL12/IL23p40 levels were significantly lower at day 3 p.i. in sera of mice infected with Δ*gra15/24* parasites compared to wild-type infected mice (**Fig 2D**), this difference was not detected in the peritoneal fluid (**Fig 2F**). On day 3 p.i., there was a large increase in IFNγ in both serum and peritoneal fluid of wild-type parasite infected mice (**Fig 2E and 2G**) which was significantly decreased in mice infected with Δ*gra15* and Δ*gra24* parasites, and even further decreased in mice infected with Δ*gra15/24* parasites (**Fig 2E and 2G**). All *Tlr11*<sup>-/-</sup> mice i.p. infected with wild-type or Δ*gra15/24* parasites died by day 10 p.i. with similar severe reduction in body weight (**S4B and S4C Fig**). Thus, GRA15 and GRA24 both contribute to induction of IFNγ from *Tlr11*<sup>-/-</sup> mice by inducing IL18 and IL12, which impacts the early intraperitoneal parasite load.

It was previously described that *Tlr11*<sup>-/-</sup> mice can survive i.p. infection with tissue cysts (20–25 cysts) [5, 6, 42]. In our hands, however, all *Tlr11*<sup>-/-</sup> mice i.p. infected with 10 cysts of wild-type or Δ*gra15/24* parasites succumbed by day 13 p.i. with similar body weight reduction, although the Δ*gra15/24* parasites caused significantly earlier mortality (**S4D and S4E Fig**). Because of the unexpected extreme susceptibility of the *Tlr11*<sup>-/-</sup> mice after i.p infection with tachyzoites or with tissue cysts, we performed s.c. infections to curb rapid *Toxoplasma* dissemination. Compared to wild-type, Δ*gra15/24* parasites caused significantly more mortality in s.c. infected *Tlr11*<sup>-/-</sup> mice (**Fig 2H**), whereas Δ*gra15* and Δ*gra15/24* parasites caused a significantly larger body weight reduction (**Fig 2I**). Thus, GRA15 and GRA24 have a significant effect on the innate immune response in *Tlr11*<sup>-/-</sup> mice. However, these mice are already extremely susceptible to wild-type parasite infection and an increased virulence of parasites without GRA15 and/or GRA24 can therefore not be detected after i.p. infection.

## The *ROP18* and *ROP5* allelic combinations determines survival of *Tlr11*<sup>-/-</sup> mice

Although there was a significant difference in cytokine induction in *Tlr11*<sup>-/-</sup> mice i.p. infected with Δ*gra15*, Δ*gra24* or Δ*gra15/24* parasites (**Fig 2B–2G**), all mice succumbed within 10–13

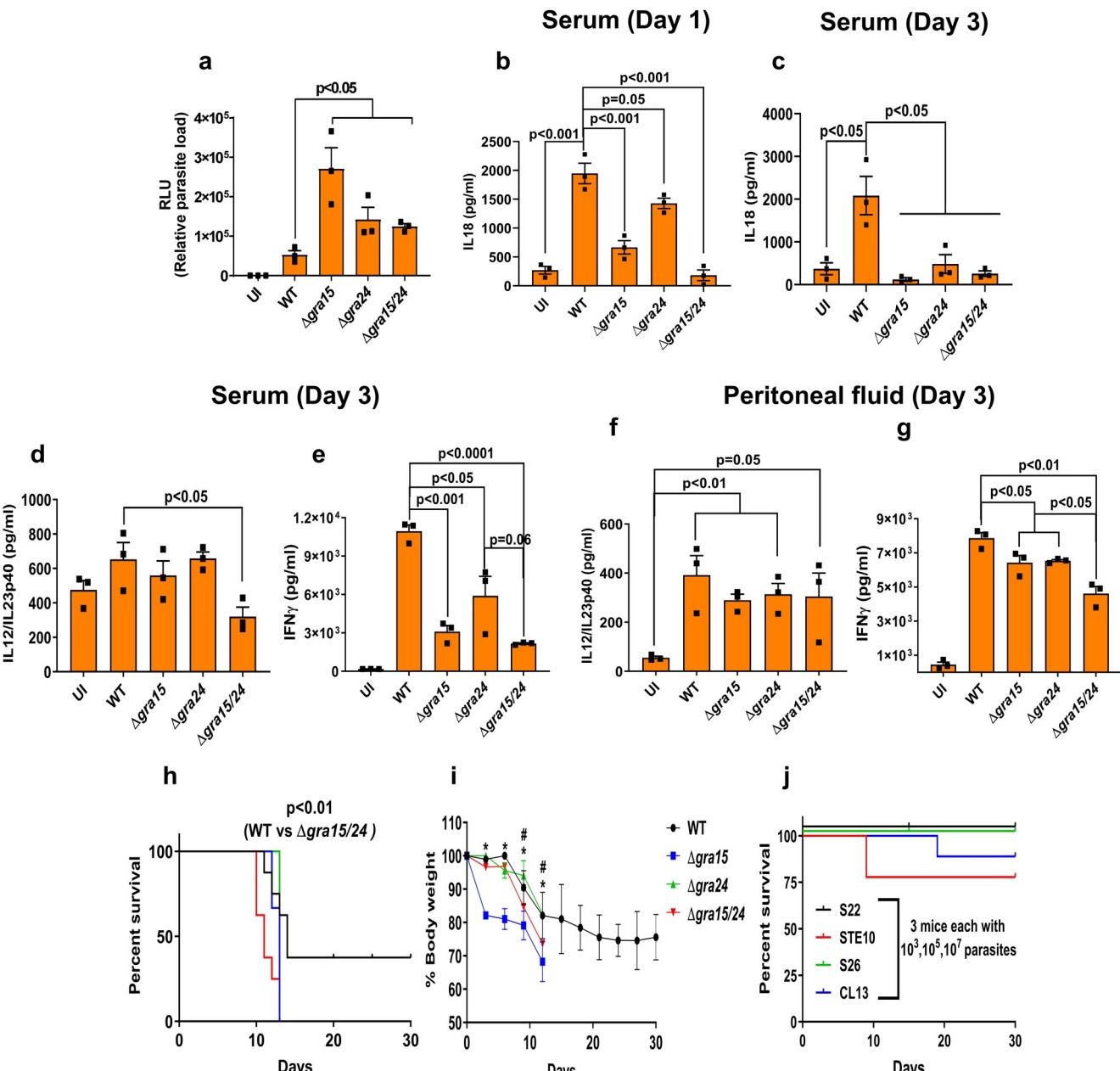

**Fig 2. Deletion of *GRA15* and *GRA24* leads to enhanced parasite virulence in *Tlr11*[-/-] mice.** *Tlr11*[-/-] mice were i.p. infected with 5,000 tachyzoites of indicated *Toxoplasma* strains expressing luciferase, and serum and peritoneal fluid were collected from each mouse (N = 3 mice per group) at different time points. (**a**) Three days p.i. peritoneal cells were isolated, plated in tissue culture plates for 24 h and the following day luciferase reading was taken to measure the parasite burden. (**b**) Serum IL18 levels at 1 day p.i. Serum IL18 (**c**), IL12/IL23p40 (**d**) and IFNγ (**e**) levels 3 days p.i. IL12/IL23p40 (**f**) and IFNγ (**g**) levels in the peritoneal fluid 3 days p.i. All data represent mean ± SEM. Statistical analysis was done with two sample Student's t tests. *Tlr11*[-/-] mice were s.c. infected with 5,000 tachyzoites of indicated strains and (**h**) survival and (**i**) weight (plotted as an average of the change in body weight for each cohort, where 100% body weight corresponds to the day of infection) of the mice were monitored for 30 days. Statistical analysis was done using a log-rank test. N = 8 each for wild-type and Δ*gra15/24* whereas N = 5 for Δ*gra15* and Δ*gra24*. *Tlr11*[-/-] mice were i.p injected with indicated doses of tachyzoites of different *Toxoplasma* strains derived from F1 progenies of a type II X type III cross (51) containing avirulent *ROP18* and *ROP5* alleles and survival was monitored (**j**). Each dot represents a mean of 3 technical replicates from an experiment. Statistical analysis was done by two sample student's t test for figures a-g. Data are represented as mean ± standard error of the mean (SEM).

days p.i. In mice, acute parasite virulence is determined by the exact *ROP18* and *ROP5* allele [10, 47–50]. To determine the role of ROP18 and ROP5 in *Tlr11*[-/-] mice we i.p. infected *Tlr11*[-/-] mice with 4 F1 progeny (S22, S26, STE10 and CL13) from a type IIxIII cross [51] that have the avirulent alleles of *ROP18* and *ROP5*. While 100% mortality was observed after infection with 100 Pru wild-type or Δ*gra15/24* parasites (**S4B Fig**), mice infected with these F1 progeny strains survived doses up to $10^5$ parasites (**Fig 2H and 2I**). Only 1 mouse infected with CL13 and 2 mice infected with STE10 parasites (out of 5) died at the $10^7$ dose (**Fig 2J**), accompanied with significant body weight reduction (7–10%) only in mice infected with the STE10 strain (**S4F–S4H Fig**). Thus, although in *Tlr11*[-/-] mice the cytokine response is significantly influenced by GRA15 and GRA24, survival after i.p. infection is almost entirely dependent on ROP18 and ROP5. It therefore appears that *Tlr11*[-/-] mice are not a good model for the human immune response to *Toxoplasma* as humans do not have IRGs and ROP18 and ROP5 do not affect *Toxoplasma* resistance to IFNγ in human cells [10, 17, 52].

## GRA15 and GRA24 induce cytokine secretion in human macrophages through activation of p38 MAPK and NFκB

The innate immune response against *Toxoplasma* in human monocytes is affected by the infecting strain type. For example, type II, but not type I strains, are major inducers of IL1β [18, 33], which is primarily dependent on type II GRA15 [33]. However, the role of GRA24 in the induction of pro-inflammatory cytokines by human macrophages is not known. We measured IL12/IL23p40, IL1β and TNFα from THP1-derived macrophages infected with wild-type, Δ*gra15*, Δ*gra24* and Δ*gra15/24* parasites and observed that, akin to murine macrophages, GRA15 and GRA24 were both important for generation of IL12/IL23p40 (**Fig 3A**). However, in contrast to murine macrophages, both GRA15 and GRA24 determined IL1β and TNFα production by THP1-derived macrophages (**Fig 3B and 3C**). This difference between human and murine macrophages could be due to species-specific transcription factor dependence as often seen specifically for macrophages [53]. We also measured the growth of the different parasite lines in THP1 macrophages and detected a growth advantage in parasites lacking *GRA24* compared to wild-type parasites (**Fig 3D**). Furthermore, similar to murine macrophages, while GRA15 and GRA24 specifically activated nuclear translocation of p65 NFκB and p38 MAPK, respectively, in THP1 macrophages (**Fig 3E and 3F**, **S5A and S5B Fig**), both GRA15 and GRA24 were required for nuclear translocation of cREL (**Fig 3G and S5C Fig**). To confirm the role of NFκB and p38 MAPK on cytokine secretion from THP1 macrophages, we inhibited the two pathways using BAY11-7082, an irreversible inhibitor of IκB phosphorylation important for NFκB activation [54] and BIRB796, a potent inhibitor of p38 MAPKα [55], respectively. Inhibition of either the NFκB or p38 MAPK pathway significantly inhibited the secretion of IL12/IL23p40 and IL1β from THP1 macrophages infected with wild-type parasites (**Fig 3H–3I**), while the inhibitors did not affect parasite growth (**Fig 3J**). Thus, GRA15 and GRA24 regulate pro-inflammatory cytokine production from THP1-derived macrophages through NFκB- and p38 MAPK-dependent pathways.

## IL12 and NLRP3 inflammasome-derived IL18 induce the secretion of IFNγ from human PBMCs

In mice, IFNγ is known to be induced by IL12, which can be further enhanced by IL18 and IL1β [14, 15]. To determine the role of IL12 and IL18 in the induction of IFNγ in human cells we used PBMCs as a model [56]. When PBMCs were infected with Pru wild-type there was a significant decrease in IFNγ secretion upon blocking IL12 or IL18 compared to either untreated or isotype antibody-treated cells (**Fig 4A**), which was more pronounced when cells

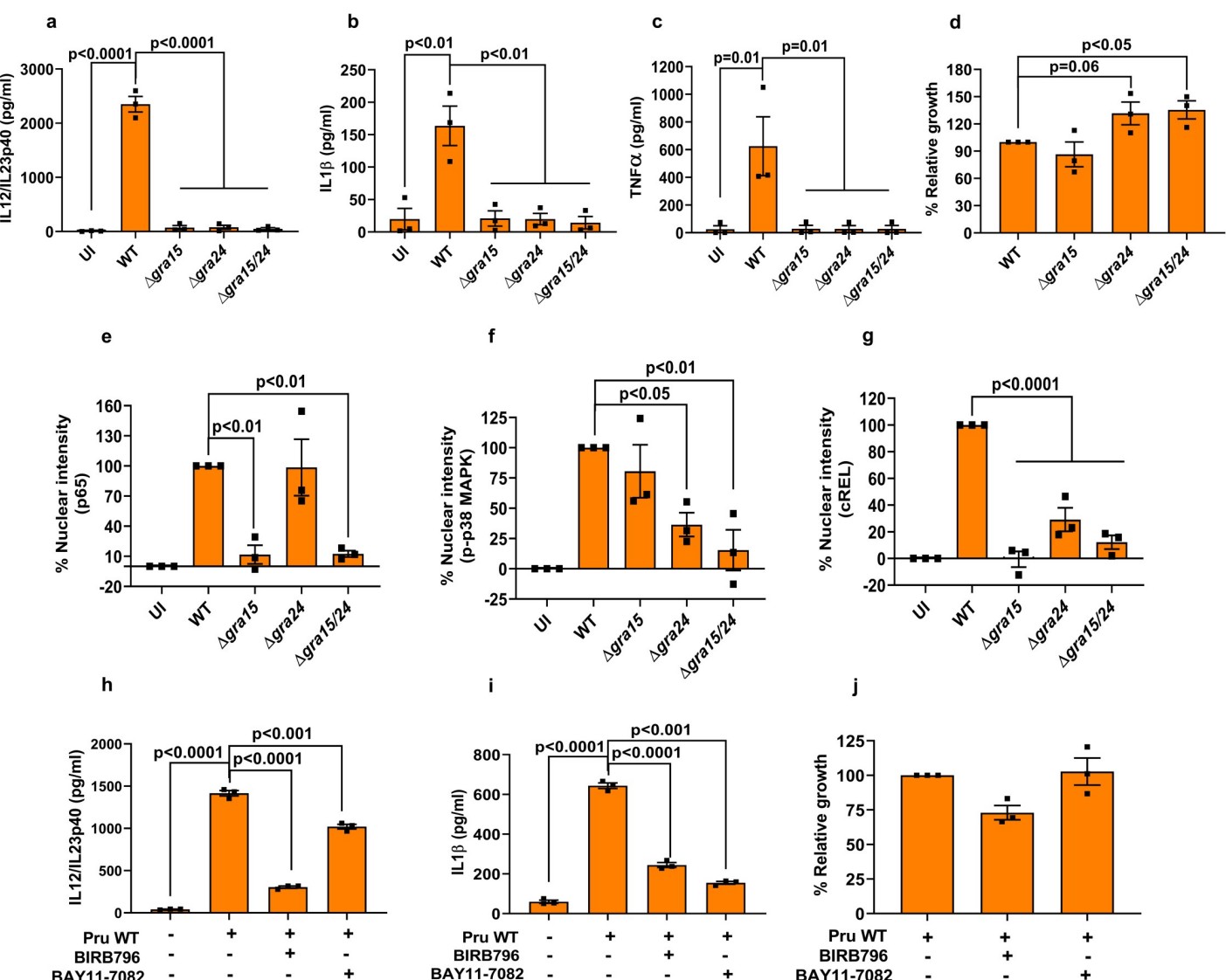

**Fig 3. GRA15 and GRA24 activate pro-inflammatory cytokine secretion by human macrophages.** THP1 monocyte-derived macrophages were infected with indicated *Toxoplasma* strains for 24 h, after which IL12/IL23p40 (**a**), IL1β (**b**) and TNFα were measured. Relative parasite growth was measured by luciferase growth assay (**d**). Nuclear translocation of the NFκB p65 (**e**), p-p38 MAPK (**f**), and NFκB cREL (**g**) subunits were quantified from infected THP1 macrophages 18 h p.i with indicated strains. In each experiment at least 15 cells were quantified. THP1 macrophages were treated with indicated inhibitors 2 h prior to infection and subsequently infected for an additional 20 h. IL12/IL23p40 (**h**), IL1β (**i**) and growth (**j**) were measured. Each dot represents a technical mean value from a single experiment, and each experiment was done 3 times. Statistical analysis was done by One-way ANOVA followed with Tukey's multiple comparison test. Data are represented as mean ± standard error of the mean (SEM).

were treated with blocking antibodies against both IL12 and IL18. These results suggest that both IL12 and IL18 are required for optimal IFNγ production by human PBMC (**Fig 4A**). This was further corroborated by the increased parasite growth detected in PBMCs treated with blocking antibodies against IL12, IL18, or both (**Fig 4B**). IL18 and IL1β are secreted as a result of inflammasome activation [12]. In *Toxoplasma*-infected human PBMCs, IL1β secretion is mediated via NLRP3 inflammasome activation [20]. Indeed, inhibition of the NLRP3 inflammasome with MCC950 or inhibition of CASP1 with VX765 led to a significant decrease in IL1β, IL18, and IFNγ secretion (**Fig 4C–4E**) accompanied by increased parasite growth (**Fig 4G**). These effects on IFNγ by the inhibitors were not dependent on IL12/IL23p40 secretion,

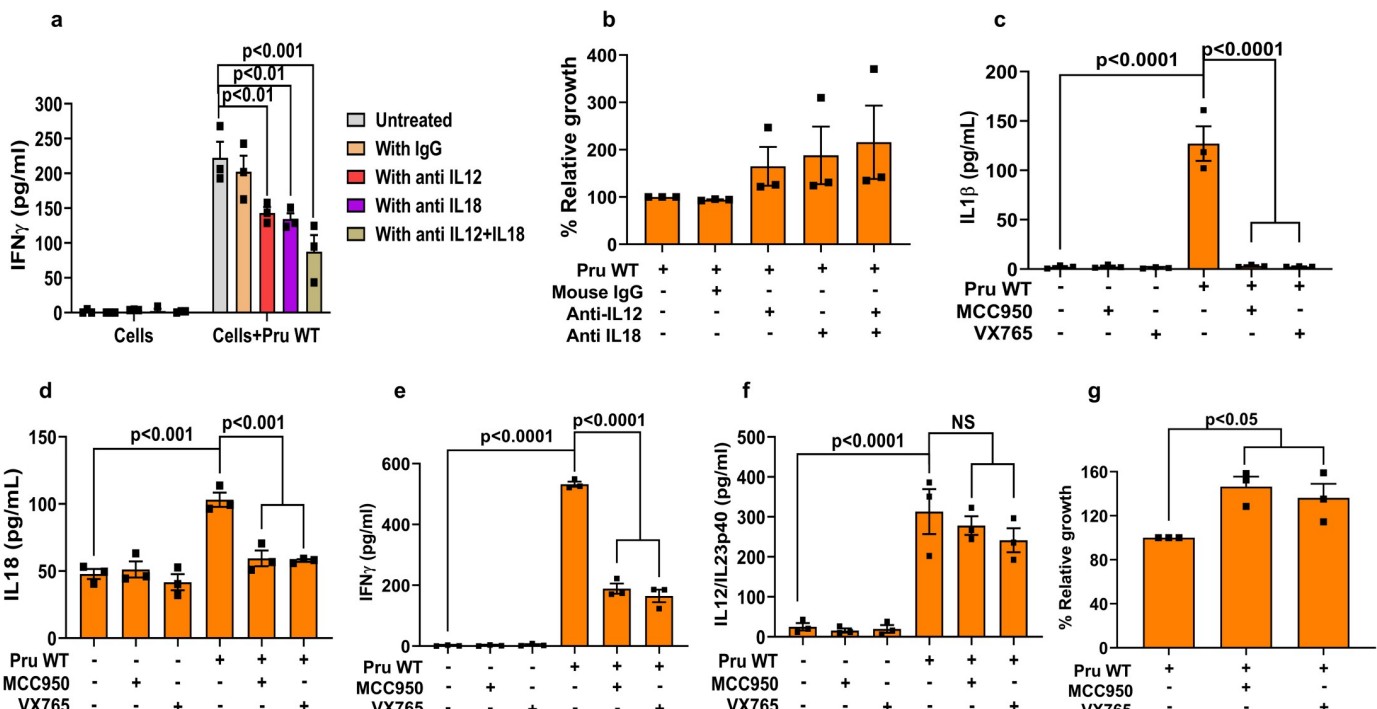

**Fig 4. Secretion of IFNγ from human PBMCs is dependent on IL12 and NLRP3 inflammasome-derived IL18 and IL1β.** PBMCs were infected with Pru wild-type parasites and treated with either anti-IL12, anti-IL18, isotype specific antibody, or anti-IL12+anti-IL18 1 h p.i. Supernatants were harvested for quantification of IFNγ (**a**) and parasite growth (**b**). PBMCs were treated with the caspase 1/4 inhibitor VX765 or NLRP3 inhibitor MCC950 2 h pre-infection followed by infection for another 20 h. After harvesting the culture supernatant, IL1β (**c**), IL18 (**d**), IFNγ (**e**) and IL12/IL23p40 (**f**) were measured. Parasite growth was measured from the cell lysate using luciferase assay (**g**). Each dot represents the mean of 3 technical replicates from a single experiment. Statistical analysis was done with Two-way ANOVA followed by Tukey's multiple comparison test (**a**), two sample Student's t test (**b and g**), and One-way ANOVA followed by Tukey's multiple comparison test (**c-f**). Data are represented as mean ± standard error of the mean (SEM).

as the inhibitors did not alter IL12/IL23p40 levels (**Fig 4F**). Thus, IFNγ secretion from human PBMCs infected with the type II Pru strain, is dependent on IL12 and NLRP3 inflammasome-derived IL18 and IL1β.

## Cytokine secretion from human PBMCs is mediated through activation of NFκB and p38 MAPK by GRA15 and GRA24

To test whether p38 MAPK and NFκB signaling are involved in cytokine production from PBMCs infected with Pru wild-type, we used the inhibitors BIRB796 (inhibits p38 MAPK) and BAY11-7082 (inhibits IκB phosphorylation). Both inhibitors inhibited secretion of IL12/IL23p40 (**Fig 5A**), IL1β (**Fig 5B**) and IFNγ (**Fig 5C**), with BAY11-7082 having a much greater effect (**Fig 5A–5C**). Furthermore, compared to untreated PBMCs, those treated with either of these inhibitors supported more parasite growth (**Fig 5D**). As we observed that GRA15 and GRA24 induced inflammatory cytokine generation in THP1-derived macrophages, we also tested their effect on human PBMCs. PBMCs infected with Δ*gra15*, Δ*gra24*, or Δ*gra15/gra24* parasites secreted significantly less IL12/IL23p40 compared to PBMCs infected with wild-type parasites (**Fig 6A**). Furthermore, parasites lacking *GRA15*, *GRA24*, or both grew more in PBMCs compared to wild type parasites probably due to lack of anti-parasitic IFNγ secretion from PBMCs infected with the knockout parasites (**S6A Fig**). Likewise, PBMCs infected with Δ*gra15*, Δ*gra24*, or Δ*gra15/gra24* parasites secreted significantly less IFNγ, TNFα and IL1β compared to PBMCs infected with wild-type (**Fig 6B–6D**). Thus, GRA15 and GRA24 together

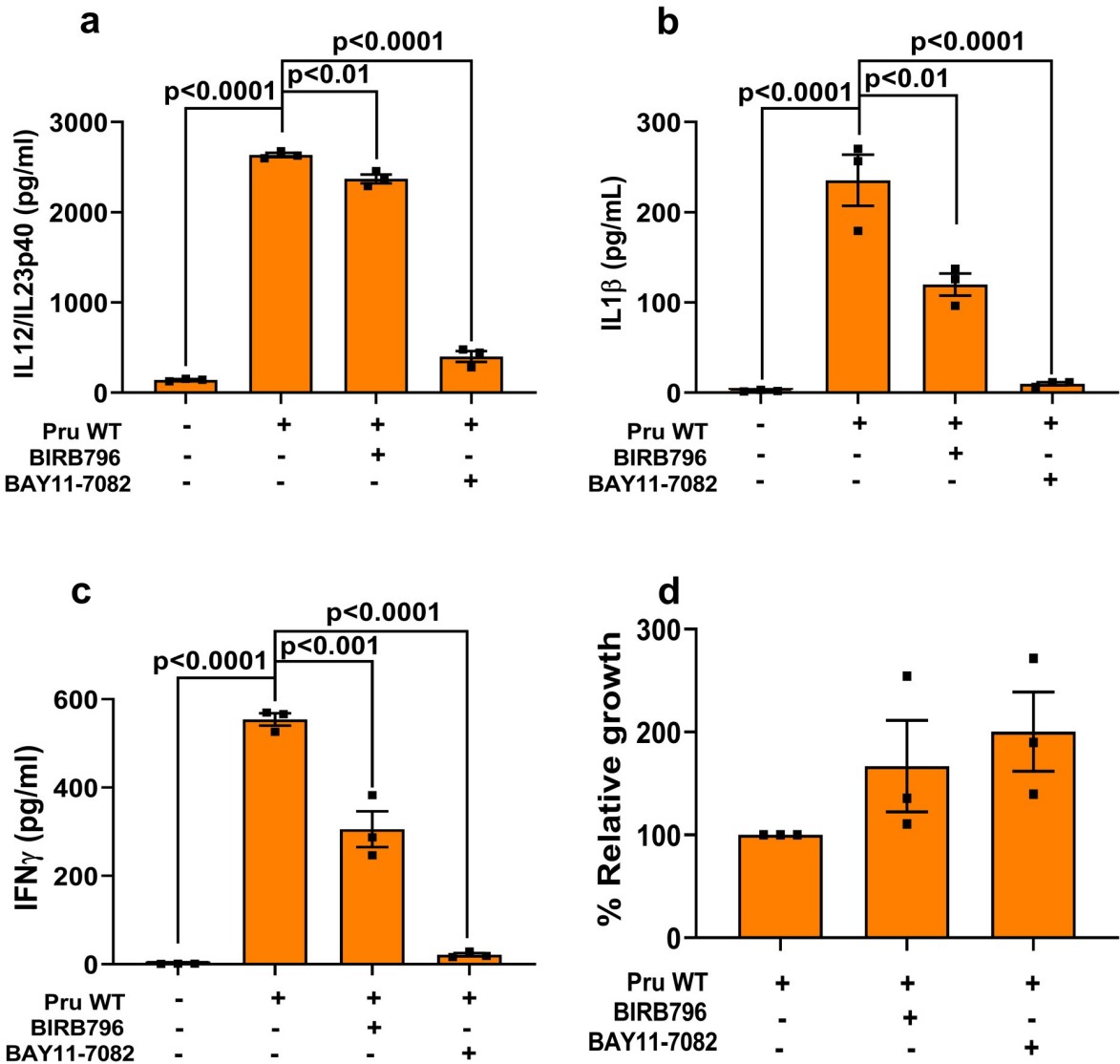

**Fig 5. Cytokine secretion from human PBMCs is mediated by activation of NFκB and p38 MAPK.** PBMCs were treated with indicated inhibitors 2 h prior to infection and subsequently infected for 20 h, after which IL12/IL23p40 (**a**), IL1β (**b**) and IFNγ were measured. The relative parasite growth was measured by luciferase growth assay (**d**). Each dot represents the mean of 3 technical replicates from an experiment. Statistical analysis was done with One-way ANOVA followed by Tukey's multiple comparison test. Data are represented as mean ± standard error of the mean (SEM).

determine the secretion of proinflammatory cytokines IL12/23p40, IFNγ, TNFα and IL1β from infected human PBMCs.

Recently, it was shown that in humans, alarmin S100A11 is released from *Toxoplasma*-infected fibroblasts and sensed by THP1 monocytes, which upregulated CCL2 production to induce recruitment of additional monocytes [18]. However, the secretion of alarmin S100A11 did not differ between uninfected and parasite infected PBMCs (**S6B Fig**). On the other hand, akin to a previous study [18], CCL2 secretion was observed from PBMCs upon *Toxoplasma* infection independent of either GRA15 or GRA24 (**S6C Fig**). Similarly, in accordance with Safronova et al. [18], HFFs infected with wild-type parasites secreted significantly more S100A11 compared to uninfected cells, which could be due to increased activity of inflammatory caspases 1 and 4 (**S6C and S6D Fig**). Secretion of S100A11 is dependent on

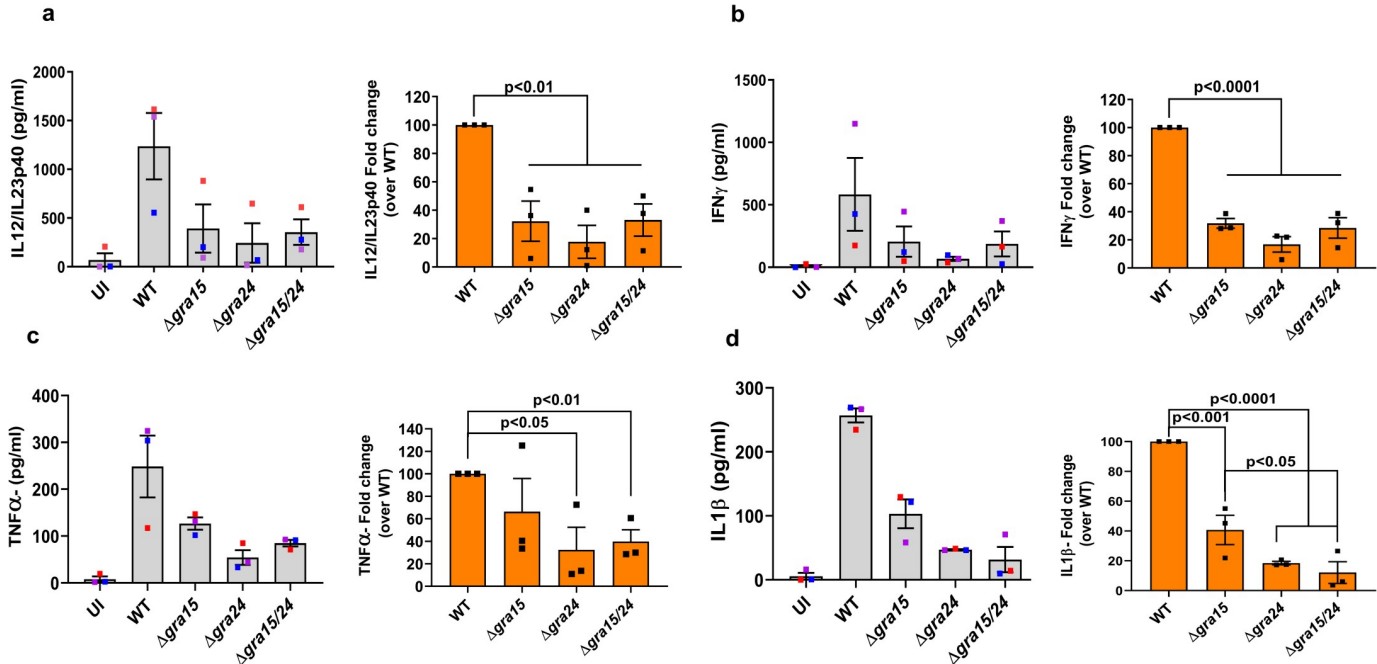

**Fig 6. GRA15 and GRA24 induce cytokine secretion by human PBMCs.** PBMCs were infected with indicated *Toxoplasma* strains for 24 h, after which IL12/IL23p40 (**a**), IFNγ (**b**), TNFα (**c**), IL1β (**d**) were measured. Each dot represents the mean of 3 technical replicates from an experiment. Statistical analysis was done with One-way ANOVA followed by Tukey's multiple comparison test. Data are represented as mean ± standard error of the mean (SEM).

permeabilization mediated cell lysis [18] and fibroblasts treated with Triton X100 released very high levels of S100A11 (**S6B Fig**). Our data indicate that inflammatory cytokine secretion from human PBMCs infected with type II *Toxoplasma* is regulated by both GRA15 and GRA24.

## Discussion

Innate recognition of *Toxoplasma gondii* DNA/RNA and profilin by nucleic acid sensing TLRs or TLR11/12 heterodimers is critical for robust IL12 production and subsequent activation of host protective IFNγ [7, 57, 58]. Nevertheless, hosts lacking TLR11/12 can still produce IL12 from monocytes and dendritic cells while IFNγ is produced from T cells, NK cells and neutrophils [18, 42, 43, 56]. Our study showed that, compared to wild-type parasite infected mice, *Tlr11*[-/-] mice infected with parasites lacking *GRA15* and/or *GRA24* have an increased parasite load which was correlated with significantly lower IL18, IL12 and IFNγ levels. However, even in the *Tlr11*[-/-] mouse model, parasite virulence is primarily determined by ROP18 and ROP5. ROP18 and ROP5 counteract the IRGs, which are not present in humans, and which likely explains why ROP18 and ROP5 do not determine parasite susceptibility to human IFNγ [10, 17, 52]. Thus *Tlr11*[-/-] mice do not appear to be a good model for the human immune response to *Toxoplasma*. We show that in human THP1-derived macrophages GRA15 and GRA24 induced IL12, TNFα and IL1β through their ability to activate the NFκB and p38 MAPK pathways. In PBMCs we show that IFNγ secretion is dependent on IL12 and NLRP3 inflammasome-derived IL18 and IL1β, which are also induced by GRA15 and GRA24. Thus, GRA15 and GRA24 are major activators of the human immune response.

In the murine model of toxoplasmosis, IL12 production is largely dependent on dendritic cells (DCs) and macrophages [4, 44, 59, 60]. However, the mechanism of IL12 production by these two cell types is different [8], as in DCs it is primarily determined by TLR11, chemokine

receptor 5 (CCR5) and the myeloid differentiation factor 88 (MyD88) pathway, with an additional role of G protein coupled receptor signaling (GPCR) [4, 61, 62]. On the other hand, generation of IL12 from macrophages is independent of TLR11 and is induced primarily by cREL NFκB driven transcription [8, 63]. Additionally, compared to cREL NFκB, p65 NFκB plays a moderate role in induction of IL12/IL23p40 [46]. The larger impact on IL12/IL23p40 production by GRA15 compared to GRA24 might be due to the activation of both p65 and cREL by GRA15, while GRA24 only activates cREL. It has been shown that p38 MAPK can phosphorylate acetyltransferase coactivator p300, which can acetylate NFκB p65 and thereby regulate its transcriptional activity [64]. Acetylated NFkB p65 can stimulate or repress the expression of specific genes [65]. Therefore, optimal activation of cREL could be mediated by both upregulation of its transcription and its nuclear translocation. This is consistent with the data from a recent study that reported GRA24-mediated mRNA induction of IL12/IL23p40 in bone marrow derived dendritic cells (BMDCs) [66]. TNFα and IL1β are regulated by the NFκB p65-p50 subunit in murine macrophages [46] which explains why these cytokines were not affected by GRA24.

Compared to an *in vitro* cell culture system, the immune response *in vivo* is much more complex, as multiple cell types interact and can exert a considerable influence on disease outcome. For instance, although *Tlr11*[-/-] mice lack the increased level of IL12 compared to wild-type mice upon *Toxoplasma* infection [4, 5, 57] the IFNγ response in these mice is intact and even higher than in wild-type mice, possibly due to the higher parasite load [42, 43, 57]. One important gap in these studies is the determination of what parasitic factors are responsible for the residual IL12 or IL18 that activate NK and T cells to produce IFNγ. Although, GRA15 and GRA24 control cytokine induction in *Tlr11*[-/-] mice, a significant level of IFNγ was still detected in *Tlr11*[-/-] mice infected with Δ*gra15/24* parasites. This is probably due to the release of parasite derived nucleic acids after IRG and GBP-mediated destruction of the PVM, which can induce interferon production through nucleic acid sensing TLRs [57, 67]. Mice lacking either *Tlr11*[-/-] or *Tlr3/7/9*[-/-] still produce IL12 and IFNγ upon *Toxoplasma* infection, whereas in 3d mice (lacking the endosomal chaperone UNC93B1 for TLR 3/7/9/11/12) or *Tlr3/7/9/11*[-/-] quadruple knockout mice, the IL12 and IFNγ response was completely abrogated [57]. This could explain why a significant level of IFNγ in *Tlr11*[-/-] mice infected with Δ*gra15/24* parasites was still detected.

It was previously shown that *Tlr11*[-/-] mice infected with ME49 tissue cysts were not more susceptible compared to wild-type mice [42, 57]. By contrast, in our study acute susceptibility of *Tlr11*[-/-] mice was observed when infected with either a low number of Pru cysts or tachyzoites which was accompanied by a steady reduction of body weights. Similarly, a recent study showed an increased susceptibility of *Tlr11*[-/-] mice after ME49 infection [43]. This variable susceptibility of *Tlr11*[-/-] mice could be due to the parasite strains used for infection, i.e ME49 *versus* Pru, which can elicit a different immune response [68]. Alternatively, the different susceptibility of *Tlr11*[-/-] mice could be explained by variations in the microbiota in these mice, as they were housed in different colonies [5, 42, 43, 57, 69]. Acute mortality of mice upon *Toxoplasma* infection depends on the route of infection and virulence of the parasite strain [70]. For virulent parasites a difference between knockout and wild-type parasites is more likely to be detected after s.c. infections, as this results in a slower dissemination of parasites from the site of infection [32]. Similarly, in our study significantly enhanced virulence of Δ*gra15/24* parasites compared to Pru wild type was only observed after s.c. infection.

The human immune response to *Toxoplasma* has been studied in THP1 monocytes [18, 33], isolated PBMCs [18], or elutriated monocytes [18, 20, 56]. Previous studies have shown that although THP1 monocytes infected with type II strains secrete IL1β [18, 33], they do not secrete IL12/IL23p40 [18]. However, we observed that PMA-differentiated THP1 macrophages

secreted both IL12/IL23p40 and IL1β in a GRA15- and GRA24-dependent way. This could be attributed to differences between monocytes used in those studies [18, 33] versus macrophages in our study as primary human monocytes and macrophages from the same donor differ in their cytokine secretion pattern [20, 56]. In the present work, we detected a large amount of IL12/IL23p40 secretion from *Toxoplasma* infected PBMCs. This is in accordance with Tosh *et al*. [56], where they showed that elutriated monocytes or column purified monocytes were equally efficient in secreting IL12/IL23p40 secretion upon *Toxoplasma* infection regardless of the strain type. On the other hand, Safronova et al. [18] did not observe any IL12/IL23p40 secretion from PBMCs infected with type II *Toxoplasma*. This discrepancy could be due to differences in the isolation and purification of monocytes, time point of the assay and, most importantly, the multiplicity of infection (MOI) [71].

THP1 cells are a homogenous monocytic cell population, whereas human PBMCs contain a mixture of cell types comprising a small proportion (approx. 10%) of monocytes [72]. T cells constitute the largest fraction of PBMCs (roughly ¾ th), and IFNγ production by these cells is primarily determined by IL12 and IL18-mediated activation of STAT4, p38 MAPK, NFκB, and activator protein 1 (AP-1) family of transcription factors, respectively [73]. It has been shown that within human PBMCs *Toxoplasma* preferentially infects monocytes but lymphocytes were also infected at a lower level [74, 75]. Furthermore, a recent study showed that another *Toxoplasma* secreted factor, '*Toxoplasma* E2F4-associated EZH2-inducing gene regulator' or TEEGR, negatively regulates the NFκB pathway and selectively suppresses the cytokines regulated by NFκB [76]. The fact that TEEGR suppresses IL1β induction but has no role in IL12 induction could indicate that TEEGR inhibits the NFκB but not the cREL pathway. Based on these facts and the results observed in our study, we hypothesize that in human PBMCs the parasite-infected monocytes produce IL12 through cREL activation and IL18 by inflammasome activation, which together activate T cells to produce anti-parasitic IFNγ which could destroy some PVs. Subsequently, PAMPs get released inside the host cytosol and could be sensed by cytosolic nucleic acid sensing TLRs such as TLR3/7/9, in turn inducing IL12. Indeed, it has been shown that IFNγ-primed human PBMCs produce IL12, IL1β and TNFα when treated with type II parasite derived DNA and RNA [57]. Taken together, we determined that the immune response against *Toxoplasma* in TLR11 deficient mice or human cells is largely dependent on GRA15/GRA24-induced inflammasome-mediated secretion of IL18/IL1β which together with IL12 activate NK and T cells to secrete IFNγ that kills the parasite. Our study advances our understanding of the human immune response against *Toxoplasma*.

## Materials and methods

### Ethics statement

Animal experiments were performed in strict accordance with the recommendations in the Guide for the Care and Use of Laboratory Animals of the National Institutes of Health and the Animal Welfare Act, approved by the Institutional Animal Care and Use Committee at UC Davis (assurance number A-3433-01).

### Culture of cells and parasites

Human foreskin fibroblasts (HFFs) and RAW 264.7 macrophages were cultured as described previously [31, 77]. All parasite lines were maintained *in vitro* by serial passage on HFFs monolayers and cultured in the same medium as HFFs but with 1% fetal bovine serum FBS). A *Toxoplasma gondii* Pru strain expressing firefly luciferase and GFP (PruΔ*hpt*, PruA7) was used as representative of type II [78].

## Generation of bone marrow-derived macrophages and THP1 macrophages

Bone marrow-derived macrophages (BMDMs) were isolated from C57BL/6 mice and cultured as described previously [13]. THP1 monocytes were cultured in RPMI-1640 supplemented with 10% FBS, 2 mM L-glutamine, 100 U/mL penicillin/streptomycin and 10 mg/mL gentamicin. For differentiation into macrophages, THP1 monocytes were stimulated with 100 nM phorbol 12-myristate 13-acetate (PMA) for 3 days and then rested for 1 day with replacement of the PMA containing medium with complete medium without PMA before performing experiments. All the experiments involving THP1 monocytes were performed with passage numbers <10.

## Generation of knockout parasites

It was previously shown that type II *Toxoplasma* strains lacking hypoxanthine-guanine phosphoribosyl transferase (*HXGPRT* or *HPT*) are more virulent than strains having *HPT* [79]. To remove the *HPT* gene from the Δ*gra15* strain, we mutated the *HPT* locus (TGME49_200320) by using a clustered regularly interspaced short palindromic repeat (CRISPR)-Cas9 based system. The sgRNA sequence against *HPT* (**Table 1**) was cloned into the pSS013-Cas9 vector (pU6 plasmid, Addgene plasmid # 52694) using the *BsaI* specific sites. To generate PruΔ-*gra15*Δ*hpt*, the circular pSS013-Cas9 vector containing the sgRNA against *HPT* was transfected (10 µg) as described elsewhere [80]. For the selection of Δ*hpt* parasites, single clones were grown in parallel with Mycophenolic acid (MPA)-Xanthine (25 µg/mL) and 6-Thioxanthine (177 µg/mL) containing media. Parasites that were able to grow in 6-Thioxanthine but not in MPA-xanthine media were selected and further confirmed by PCR and sequencing (**Table 1**). To disrupt *GRA24* (TgME49_230180) in PruΔ*hpt* and PruΔ*hpt*Δ*gra15* strains, the pSS013-Cas9 vector containing a sgRNA against *GRA24* was transfected along with *NotI* (New

**Table 1. Primer and sgRNA sequences.**

| Name | Sequence |
|---|---|
| TGME49_200320_gRNA1_Fwd | 5' AAGTT **GACAAAATCCTCCTCCCTGG** G 3' |
| TGME49_200320_gRNA1_Rev | 5' AAAAC **CCAGGGAGGAGGATTTTGTC** A 3' |
| TGME49_200320_gRNA2_Fwd | 5' AAGTT **GGACATAGTGCTCGAAGAAG** G 3' |
| TGME49_200320_gRNA2_Rev | 5' AAAAC **CTTCTTCGAGCACTATGTCC** A 3 |
| TGME49_230180_gRNA1_Fwd | 5' AAGTT **GTACCAGGCTACAAATAGAGA** G 3' |
| TGME49_230180_gRNA1_Rev | 5' AAAAC **TCTCTATTTGTAGCCTGGTAC** A 3' |
| TGME49_230180_gRNA2_Fwd | 5' AAGTT **GGGACCGAAATGCCGAATCA** G 3' |
| TGME49_230180_gRNA2_Rev | 5' AAAAC **TGATTCGGCATTTCGGTCCC** A 3 |
| HPT_Fwd | 5' ATGGCGTCCAAACCCATTGA 3' |
| HPT_Rev | 5' TCGTTGAAGTCGTAGCAGCA 3' |
| GRA24_Fwd | 5' ATGCTCCAGATGGCACGATATACCG 3' |
| GRA24_Rev | 5' CTGTCGTCTGCTGGTGGTAGC 3' |
| DHFR_Rev | 5' ATAGTCCTGTCGGGTTTCGCCAC 3' |
| GRA15_Fwd | 5' AACACGACGAGGCAGGAGAATTAC 3' |
| GRA15_Rev | 5' GACGACTGTAGCCTGAGCATCC 3' |
| Neo74 | 5' GTGGGATTAGATAAATGCCTGCTC 3' |
| 5ARMF2 | 5' AACACAGGCTCAGAAGAGAAGAGG 3' |
| 2285 | 5' TTGATGTATTCGTGTCCCACTGC 3' |
| +3AMR | 5' GCGGACACCTTCCATCTCTCAGTT 3' |

Nucleotides highlighted in grey are part of the *BsaI* cloning site of pSS013 and nucleotides in bold are the actual sgDNA sequence.

England Biolabs) linearized pLoxP-DHFR-mCherry-LoxP (Addgene Plasmid #70147), at a 5:1 molar ratio, as described previously [81]. This plasmid contains a pyrimethamine resistance cassette tagged with the fluorescence marker mCherry and flanked by two LoxP sites. After three rounds of pyrimethamine selection (1 μM) and limiting dilution cloning, *GRA24* knockout parasites were assessed by PCR and confirmed by sequencing (**Table 1**). To flox out DHFR-mCherry, Δ*gra24* and Δ*gra15/24* parasites were transfected with a plasmid [78] expressing Cre recombinase (50 μg) as described above. Single clones were checked for their inability to grow in the presence of pyrimethamine and absence of mCherry.

### Generation and maintenance of *Tlr11*$^{-/-}$ mice

To generate the *Tlr11*$^{-/-}$ mouse colony, two 4-week-old *Tlr11*$^{-/-}$ male mice [42, 69] were bred with wild-type female C57BL/6 mice (Jackson laboratories). The *Tlr11*$^{+/-}$ F1 progeny mice were subsequently crossed to obtain *Tlr11*$^{-/-}$ mice. Genotyping of F2 individuals was performed by PCR from DNA isolated from tail clips to identify the hybrid or homozygous variants (**S2 Fig**). Once the homozygous mice were confirmed by PCR genotyping (**S2 Fig**), they were bred to get the entire colony of *Tlr11*$^{-/-}$ mice. Mice were maintained at the University of California, Davis (UC Davis) mouse housing facility, where water and feed were provided *ad libitum*.

### *In vivo* infection, parasite burden and cytokine measurement

Male and female 6–10-week-old *Tlr11*$^{-/-}$ mice were used in the experiments. For infection, tachyzoites were cultured in HFFs and extracted from host cells by passage through 27- and 30-gauge needles, washed two times in PBS, and quantified with a hemocytometer. Parasites were diluted in PBS, and mice were inoculated either i.p. or s.c. with tachyzoites of each strain (100–5,000 tachyzoites in 200 μl) using a 29-gauge needle. Body weights were recorded every day and for survival analysis mice were kept for 30 days. To obtain brain cysts, 5,000 tachyzoites of wild-type and Δ*gra15/24* parasites were injected i.p. into CD1 mice (Charles River) and 4 weeks later these mice were sacrificed, brains aseptically collected and cysts isolated for both strains [41]. Subsequently, 10 cysts of each parasite strain were infected i.p. into *Tlr11*$^{-/-}$ mice to determine survival and body weight reduction.

*In vivo* parasite burden was measured by i.p. infecting 5,000 tachyzoites into *Tlr11*$^{-/-}$ mice for 3 days. Subsequently, peritoneal fluids were collected and cells isolated by centrifugation. A total of $1\times10^5$ cells were plated in 96-well plates in triplicate for each group for 24 h. Following incubation, supernatants were removed and lysis buffer was added prior to measure relative parasite growth by luciferase assay [77].

To quantify cytokines *in vivo*, mice were sacrificed on day 1 or 3 p.i. to collect blood and peritoneal fluid. Serum was diluted 1:10 for IL12/IL23p40 and IL18, and 1:20 for IFNγ and TNFα measurement. IFNγ, IL12/IL23p40, TNFα and IL18 levels were determined using commercially available matched pair ELISA kits (Invitrogen, Thermo Fisher Scientific and Sino Biologicals for IL18), following the manufacturer's instructions.

### *In vitro* parasite growth determination

Freshly confluent HFFs 24-well plates were used to determine relative parasite growth by plaque assay. On the day of infection, the media was replaced, and 250 freshly harvested parasites were added to each well. Plates were then left undisturbed for 6 days at 37˚C 5% $CO_2$, after which plaque areas were imaged and measured. Plaque areas were captured and analyzed using a Nikon TE2000 inverted microscope equipped with Hamamatsu ORCA-ER digital camera and NIS Elements Imaging Software, respectively. For all experiments, at least 20–25

plaques from technical duplicate wells were imaged. For measurement of total parasite growth, a luciferase-based assay was performed [77].

## Immunofluorescence detection of p65 (NFκB), p-p38 MAPK and c-REL (NFκB) nuclear translocation

Immunofluorescence to detect nuclear translocation of p65 (NFκB), p-p38 MAPK and c-REL was done in HFFs, MEFs, RAW 264.7 macrophages and THP1-derived macrophages using the following antibodies: rabbit anti p65 (1:200 dilution, sc-109, Santacruz Biotechnology, CA, USA), rabbit anti p-p38 MAPK (1:800 dilution, #4511, Cell Signaling Technology, MA, USA), rabbit anti cREL (1:500 dilution, #4727, Cell Signaling Technology, MA, USA) and mouse anti cREL (1:200 dilution, NBP2-37593, Novus Biologicals, CO, USA for RAW 264.7 macrophages). Briefly, cells were plated on coverslips in 24-well plates ($1\times10^5$ cells/well) and subsequently infected with *Toxoplasma* with a MOI of 3 for 24 h. Following incubation, cells were fixed with 3% formaldehyde, permeabilized and blocked with buffer containing 0.2% Triton X-100 along with 3% BSA and 5% goat serum. Cells were incubated with primary antibodies overnight at 4˚C, after which each well was washed 3 times with 1×PBS, followed by incubation with goat anti rabbit Alexa fluor 594 (1:1,000 dilution, Invitrogen) and Hoechst 33258 (1:500 dilution, Invitrogen, Thermo Fisher Scientific) for 1 h. Finally, coverslips were washed 5 times with 1×PBS and mounted with VECTASHIELD antifade mounting medium (Vector Laboratories, CA, USA). Nuclear intensity of at least 15 infected cells was measured for each experiment and coverslip. Pictures were taken using NIS-Elements software (Nikon) and a digital camera (Cool SNAP EZ; Roper Scientific) connected to a fluorescence microscope (model eclipse Ti-S; Nikon). Mean fluorescence intensity quantification of nuclear signal was performed using the NIS-Elements software and Hoechst dye to define nuclei.

## Isolation of human peripheral blood mononuclear cells (PBMCs)

PBMCs were isolated from leukocyte reduction chambers (LRS) from individual donors, which were purchased from BloodSource (CA, USA) and tested seronegative for *Toxoplasma*. After collecting the blood from the LRS according to the manufacturer's protocol using sterile needles and blades, PBMCs were isolated using ficoll-Paque premium 1.077 gm/dL (GE Healthcare, PA, USA) as described previously [82]. Isolated PBMCs were subsequently used for experiments using RPMI-1640 supplemented with 10% FBS, 2 mM L-glutamine, 100 U/mL penicillin/streptomycin and 10 mg/mL gentamicin or kept frozen in 90% FBS and 10% dimethyl sulfoxide (DMSO) for later use.

### *In vitro* cytokine ELISA

C57BL/6 BMDMs, Raw 264.7 macrophages, THP1 macrophages or PBMCs were seeded ($1\times10^5$ cells per well) in 96-well plates at 37˚C in 5% $CO_2$. Cells were infected with freshly lysed tachyzoites of the different parasite strains at MOI = 3, 5 and 7, and supernatants (200 μl) were collected 24 h p.i. and used to determine IL12/IL23p40, IL1β, TNFα, IL18 and IFNγ levels. To verify that cells were infected with equal numbers of viable parasites, plaque assays were performed as described above. All the cytokine levels were measured using commercially available matched pair ELISA kits for both mouse and human (Invitrogen, Thermo Fisher Scientific), following the manufacturer's instructions.

In some assays, THP1 cells or PBMCs were first treated 2 h prior to infection with different inhibitors: BAY 11–7082 at 5 μM, (APExBIO, TX, USA), BIRB796 at 10 μM (Tocris Bioscience, MN, USA), VX765 at 50 μM (Sellelckchem, TX, USA) and MCC950 at 10 μM (Adipogen Life Sciences, CA, USA), and culture supernatants collected after 20 h. A total of 30 μL of cell

lysis buffer was added per well and a luciferase-based growth assay was performed as mentioned above.

For IFNγ determination in IL12 and/or IL18 neutralized conditions, PBMCs were first infected with Pru wild-type parasites and 1 h p.i. cells were treated with either 2 μg/mL of IL12, IL18 or isotype specific antibodies (MBL International, Japan), or both IL12 and IL18 together for another 20 h before harvesting the culture supernatant. Parasite growth was measured by luciferase assay as described above.

## Measurement of alarmin S100A11 by ELISA

Alarmin S100A11 was measured by using a commercially available pre-coated kit from Ray-Biotech (GA, USA). Briefly, PBMCs ($1 \times 10^5$ cells/well) or HFFs ($2 \times 10^4$ cells/well) were seeded in 96-well plates at 37˚C in 5% $CO_2$ and subsequently infected with freshly lysed tachyzoites of the different parasite strains at MOI = 3, 5 and 7, and supernatants (100 μl) were collected 24 h p.i. As a positive control for S100A11 secretion, 2% Triton X-100 was used in complete medium. To compare the level of S100A11 secretion between different strains, parallel plaque assays were performed.

## Arginase assay from macrophages

Arginase activity was measured from lysates of RAW 264.7 macrophages infected with different strains of *Toxoplasma* 24 h p.i in a 96-well plate with three different MOIs each time as described previously [77]. To determine that cells were infected with equal numbers of viable parasites, plaque assays were performed as described above and values relativized.

## Caspase 1/4/5 activity assay

Activity of the inflammatory caspases 1/4/5 was measured using a commercially available Caspase-Glo 1 Inflammasome Assay (Promega, WI, USA) from HFFs seeded in 96-well plates ($2 \times 10^4$ cells per well). HFFs were pre-treated with an inhibitor of caspases 1/4/5 (VX765 at 50 μM) 2 h before infection and subsequently infected with Pru wild-type parasites for another 20 h. Caspase activity was measured from the lysate and data were recorded from a single channel luminometer with a 10 s delay program.

## Supporting information

**S1 Fig. Generation of knockout strains.** Sequence of the Hypoxanthine-guanine phosphoribosyl transferase (*HPT*) gene showing the sgDNA sequence in red for Cas9-mediated disruption and primer sequence in yellow (**a**). Disruption of the *HPT gene* was determined by using specific primers designed to amplify the region shown in the bottom figure, while *GRA27* was used as a housekeeping PCR control (at the top) (**b**). Schematic diagram of the strategy followed to delete *GRA24* (top) and PCR to screen the clones confirming the disruption of the gene (P1+P2) (bottom) is shown in (**c**). wild-type (left) and the presence of the insertion of the repair template in the locus (P1+P3) (right). Identification of Δ*gra24* and Δ*gra15/24* double knockout using specific primer sets for *GRA24* (top panel), *GRA15* (middle panel) and *GRA27* as a control for quality of the input DNA (lower panel) (**d**). Nuclear translocation of the p65 subunit of NFκB was quantified from infected RAW 264.7 macrophages 18 h p.i. with indicated strains. At least 15 cells were quantified as shown in the graph (left) and representative images are shown on the right (**e**). Phenotypic confirmation of single clones of wild-type, Δ*gra15*, Δ*gra24* and Δ*gra15/24* parasites by their ability to activate NFκB (**f,g**) and p38 MAPK (**h**). Each dot represents the mean value of at least 15 host cell nuclei (**f and h**) or 3 technical

replicates (**g**) from a single experiment. Statistical analysis was done by One-way ANOVA followed by Tukey's multiple comparison test. Data are represented as mean ± standard error of the mean (SEM).
(TIF)

**S2 Fig.** Mouse breeding scheme to generate in house *TLR11* knockout mice by cross-breeding homozygous *Tlr11*-/- male with homozygous *Tlr11*+/+ female mice (**a**). Primers in the TLR11 locus (top) and PCR of the F1 progeny (bottom) where the homozygous *Tlr11*+/+ yields a single band around 700 bp, homozygous *Tlr11*-/- generates a single band around 900 bp and all heterozygous mice generate both the bands at 700 bp and 900 bp (**b**).
(TIF)

**S3 Fig. Effect of GRA15 and GRA24 on cREL nuclear translocation.** Indicated parasite strains were added (MOI of 3) to confluent monolayers of HFFs grown on coverslips in 24-well plates. 16 h p.i. cells were fixed and stained with cREL antibody. Each dot represents the mean value of at least 15 host cell nuclei from a single experiment. A representative image for each group is shown on the right. Scale bar represents 10 μm. All the data are shown as mean ± SEM. Statistical analysis was done by two sample Student's t test.
(TIF)

**S4 Fig.** *Tlr11*-/- mice were i.p infected with 5,000 tachyzoites of indicated *Toxoplasma* strains and 1-day p.i. serum was collected from each of the groups to measure IL12/IL23p40 (**a**). Survival and body weight measurements of *Tlr11*-/- mice (N = 8 mice per group) that were i.p infected with 100–1000 tachyzoites (**b-c**) or 10 tissue cysts (**d-e**) of the indicated strains. *Tlr11*-/- mice were i.p injected with indicated doses of tachyzoites of different *Toxoplasma* strains derived from F1 progenies of type II X type III crosses (51) and body weight was measured daily throughout the infection (**f-h**). All the data are represented as mean ± SEM. Statistical analysis was done by two sample Student's t test and log rank test for survival curve.
(TIF)

**S5 Fig.** PMA differentiated THP1 macrophages were infected with indicated strains for 24 h and immunofluorescence assay was performed to quantify nuclear translocation of the NFκB p65 subunit (**a**), p-p38 MAPK (**b**) and NFκB cREL subunit (**c**). Scale bar represents 10 μm.
(TIF)

**S6 Fig.** PBMCs or HFFs were infected with indicated *Toxoplasma* strains at three different MOIs for 24 h, after which supernatants were collected to measure S100A11 in PBMCs (**a**) and the PBMC lysates were used to measure parasite growth (**b**). CCL2 was measured from culture supernatants of PBMCs infected with indicated strains as described above (**c**). S100A11 was measured in HFFs (**d**). Caspase 1/4 activity assay was measured from HFFs as described in materials and methods (**e**). Each dot represents the mean value of 3 technical replicates performed for each experiment. Statistical analysis was performed by One-way ANOVA followed by Tukey's multiple comparison test. Data are represented as mean ± standard error of the mean (SEM).
(TIF)

## Acknowledgments

Authors are thankful to Professor Andreas J. Bäumler and Professor Felix O. Yarovinsky for providing the pair of male *Tlr11*-/- mice and all the members of Saeij lab for providing meaningful insights.

## Author Contributions

**Conceptualization:** Debanjan Mukhopadhyay, Jeroen P. J. Saeij.

**Data curation:** Debanjan Mukhopadhyay.

**Formal analysis:** Debanjan Mukhopadhyay, David Arranz-Solís.

**Funding acquisition:** Debanjan Mukhopadhyay, Jeroen P. J. Saeij.

**Investigation:** Debanjan Mukhopadhyay, David Arranz-Solís, Jeroen P. J. Saeij.

**Methodology:** Debanjan Mukhopadhyay, David Arranz-Solís.

**Project administration:** Jeroen P. J. Saeij.

**Supervision:** Jeroen P. J. Saeij.

**Validation:** Debanjan Mukhopadhyay.

**Visualization:** Debanjan Mukhopadhyay.

**Writing – original draft:** Debanjan Mukhopadhyay, Jeroen P. J. Saeij.

**Writing – review & editing:** Debanjan Mukhopadhyay, David Arranz-Solís, Jeroen P. J. Saeij.

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
