## [Decision Letter · Decision Letter 0]

10 Apr 2020

Dear Dr. Saeij,

Thank you very much for submitting your manuscript "Toxoplasma GRA15 and GRA24 are important activators of the host innate immune response in the absence of TLR11" for consideration at PLOS Pathogens. As with all papers reviewed by the journal, your manuscript was reviewed by members of the editorial board and by several independent reviewers. In light of the reviews (below this email), we would like to invite the resubmission of a significantly-revised version that takes into account the reviewers' comments.

We cannot make any decision about publication until we have seen the revised manuscript and your response to the reviewers' comments. Your revised manuscript is also likely to be sent to reviewers for further evaluation.

Sincerely,

Ricardo T Gazzinelli

Associate Editor

PLOS Pathogens

Vern Carruthers

Section Editor

PLOS Pathogens

Kasturi Haldar

Editor-in-Chief

PLOS Pathogens

orcid.org/0000-0001-5065-158X

Michael Malim

Editor-in-Chief

PLOS Pathogens

orcid.org/0000-0002-7699-2064

Reviewer's Responses to Questions

**Part I - Summary**

Reviewer #1: In mice, tlr11-mediated responses are of uttermost importance in the response against the parasite, meaning that more subtle contributions of other pathways are often masked by tlr11. This is circumvented using tlr11 knockout animals. Here, the authors compare WT and Δgra15/24 parasites regarding the immune responses elicited in tlr11 knockout mice, and apply the knowledge gained to systems containing human cells, who do not express tlr11.

In mice, when profilin responses are taken out of the system, GRA15 and GRA24 responses become clearer, but still of secondary relevance when compared to the responses mediated by ROP18 and ROP5. This limits the use of tlr11 knockout mice as a tool to compare mice and humans in this infection model, as IRGs play a major role in mice and not in humans. This limitation is, however, acknowledged in the manuscript, and the conclusions taken from the experiments in tlr11 ko mice are largely validated in human cells, most notably PBMCs.

Here, data points to an essential role of both GRA15 and GRA24 in eliciting IL-1, IL-12, and consequently IFN-g, in PBMCs, linking those cytokines to their transcriptional factors in both models. The study sheds light on the innate immune response elicited by the host upon Toxoplasma gondii infection, enhancing our understanding on how GRA15 and GRA24 trigger an immune response in humans.

Reviewer #2: The following manuscript brought strong arguments that complete previously published data on how GRA15 teams up with the exported effector GRA24 to induce IL-12, TNFα and IL-1β through their ability to activate the NFκB and p38alpha MAPK pathways. TLR11/12 have been reported to mediate important responses for the mitigation of T. gondii infection (Yarovinsky et al., 2005; Koblansky et al., 2013). TLR11- and TLR12-deficient mice were reported to have an impaired IL-12 response to T. gondii linked to the ability to sense the parasite-derived profilin (Yarovinsky et al., 2005; Andrade et al., 2013; Plattner et al., 2008). The authors reported here that in Tlr11-/- mice and in human cells, which do not have TLR11, the T. gondii GRA15 and GRA24 effectors play an important role in induction of IL-12, IL-18 and IL-1β, and thus in the subsequent protective IFNγ secretion. Overall, the data is of high quality, with good controls and the manuscript is well written.

**Part II – Major Issues: Key Experiments Required for Acceptance**

Reviewer #1: 1) The authors presented data throughout the paper using technical replicates. Whilst I don’t think this is necessarily wrong, I suggest the authors to make it explicit (in the figure legends) the number of independent experiments done in each case. If no other biological replicate was performed, this should also be written unambiguously.

2) Quantification of c-Rel, p65, and MAPK translocation (figures 1, 3, S1, S3, S5) does agree with the ELISA data, and tell an interesting story. I do, however, have some concerns regarding the data, making it, in my view, necessary to perform the analysis again.

- The quantitation performed in the translocation experiments used an extremely small number of cells. Variability at the single-cell level is notoriously high, as are the variabilities in the quantification of biological data from microscopic images. I recommend increasing the number of cells quantified in each group. Similar experiments in the literature usually perform this analysis in >100 randomly selected cells per treatment.

- The nuclear fluorescence/total fluorescence ratio (or perhaps the ratio between nuclear fluorescence/cytoplasmic fluorescence) would be more accurate, as this would take into account factors such as differences in fluorescence intensity between different samples, the expression levels of c-Rel/p65 in individual cells, etc.

3) Even though Alarmin levels were similar across the board, I wonder whether the effects of GRA15 and/or GRA24 on other pathways could still influence CCL2 production in PBMCs. Performing CCL2 elisas could further strengthen conclusions on this topic.

Reviewer #2: The images (Fig. 1e) of c-REL translocation are not very convincing. Authors are encouraged to submit more compelling images, if possible. Is also c-REL protein induced by T. gondii infection (as the pictures in Fig. 1e would tend to show)? therefore, is the effect mediated by GRA15 and GRA24 on (only) c-REL translocation (what the authors measure) or likely on its expression; in this regards, c-REL, unlike p65, would belong to the second wave of genes regulated by T. gondii infection. This could change the model of how GRA15 and GRA24 team up, knowing that they have distinct location in the infected cell and likely different kinetics of secretion.

**Part III – Minor Issues: Editorial and Data Presentation Modifications**

Reviewer #1: Regarding the translocation experiments, please include in the methodology information on how the quantification was performed (software, plugins, etc.).

On a cosmetic note, it is hard to visualise some of the translocation events presented, due overlap of blue (which looks saturated in some figures) and red. Addition of the isolated red and blue channels would help.

Some of the ELISA data presented in the article (notably, IL-12 in SF4-a) shows quantification of really small amounts of cytokines. Could the authors amend the methods section to make it more clear which ELISA kits were used for each cytokine?

Reviewer #2: 1- TLR11 and TLR12 both contribute to IL-12 production by macrophages, but also conventional DCs and pDCs (Raetz et al., 2013; Koblansky et al., 2013). A recent paper described how IL-12 and IL-2 responses of Toxoplasma-challenged DCs were modulated in a GRA24-dependent fashion (Ten Hoeve AL et al., Front Cell Infect Microbiol. 2019, PMID: 31681626). These data should be discussed knowing the major role DCs are playing in response to T. gondii infections and specifically IL-12 secretion in vivo.

2- Lines 212-216: the authors showed that Tlr11-/- mice infected by i.p. route with 10 cysts of Toxoplasma succumbed to acute toxoplasmosis regardless of the genotype of the strains, refuting previous studies showing that Tlr11-deficient mice can survive i.p. infection with tissue cysts (Koblansky AA et al. 2013; Raetz M et al., 2013; Sturge CR et al. 2013). IP delivery of cysts does not reflect the natural route of infection which is oral ingestion of cysts. It is quite arduous to control the effective load of infectious cysts when these are injected by the IP route. Therefore, these data can be removed to keep only infections with tachyzoite, recognized as relevant in the community.

3- The IL12/IL23p40 production is as reported here mainly controlled by GRA15 as a consequence of activation of both NF-kB p65 and c-REL subunits by GRA15, while GRA24 exclusively activates c-REL. T. gondii is also able to counteract the NF-kB pathway using exported effector(s) such as TEEGR (Braun L, et al. Nature Microbiology, 2019, PMID: 31036909). The discussion must be widened to integrate this important component, especially when we know the peculiar role of TEEGR that selectively represses the transcription of a subset of NF-κB-regulated cytokines (IL-1β, IL-6, IL-23A and IL-15) and chemokines (IL-8 and CCL20) without altering the expression of other NF-κB-regulated cytokines (IL-12 and IL-18).

PLOS authors have the option to publish the peer review history of their article (what does this mean?). If published, this will include your full peer review and any attached files.

Reviewer #1: No

Reviewer #2: Yes: HAKIMI Mohamed-Ali
---

## [Decision Letter · Decision Letter 1]

30 Apr 2020

Dear Dr. Saeij,

We are pleased to inform you that your manuscript 'Toxoplasma GRA15 and GRA24 are important activators of the host innate immune response in the absence of TLR11' has been provisionally accepted for publication in PLOS Pathogens.

Best regards,

Ricardo T Gazzinelli

Associate Editor

PLOS Pathogens

Vern Carruthers

Section Editor

PLOS Pathogens

Kasturi Haldar

Editor-in-Chief

PLOS Pathogens

orcid.org/0000-0001-5065-158X

Michael Malim

Editor-in-Chief

PLOS Pathogens

orcid.org/0000-0002-7699-2064

Reviewer Comments (if any, and for reference):

Reviewer's Responses to Questions

**Part I - Summary**

Reviewer #1: The study further consolidates the links between T. gondii GRA15/24 and the induction of cytokines such as IL-12 is humans and mice lacking tlr-11. The differences between the revised version and the original manuscript are fairly small (as it should be the case, since the original manuscript was already of high quality).

I appreciate the clarification regarding the number of biological replicates and the inclusion of CCL2 data. Regarding the quantification of c-Rel translocation, I agree that having the mean values of each group (n=3) is preferable, but I still believe the number of cells quantified is on the low side. Despite this shortcoming, I think the way it is presented is acceptable, given that the results agree with previous data as pointed out by the authors, and that performing this experiment in the current lockdown is not viable. The inclusion of the individual channels in the microscopy images is also appreciated.

Reviewer #2: This is a timely and well-done study. The authors responded to all the points this reviewer raised. The paper was significantly improved with the addition of new controls and complementary experiences.

**Part II – Major Issues: Key Experiments Required for Acceptance**

Reviewer #1: (No Response)

Reviewer #2: (No Response)

**Part III – Minor Issues: Editorial and Data Presentation Modifications**

Reviewer #1: (No Response)

Reviewer #2: (No Response)

PLOS authors have the option to publish the peer review history of their article (what does this mean?). If published, this will include your full peer review and any attached files.

Reviewer #1: No

Reviewer #2: Yes: Mohamed-Ali HAKIMI

---

## [Editor Report · Acceptance letter]

15 May 2020

Dear Dr. Saeij,

We are delighted to inform you that your manuscript, "Toxoplasma GRA15 and GRA24 are important activators of the host innate immune response in the absence of TLR11," has been formally accepted for publication in PLOS Pathogens.

Best regards,

Kasturi Haldar

Editor-in-Chief

PLOS Pathogens

orcid.org/0000-0001-5065-158X

Michael Malim

Editor-in-Chief

PLOS Pathogens

orcid.org/0000-0002-7699-2064